# OpenReview forum: "PSBench: Editing Image via  GUI Agents in Photoshop"
_ICLR.cc/2026/Conference — Submitted to ICLR 2026_

### Official Review · Reviewer_TMYi · 2025-10-25

**Soundness:** 2
**Presentation:** 2
**Contribution:** 3
**Rating:** 4
**Confidence:** 2

**Summary:**

This paper introduces PSBench, a benchmark for evaluating the Photoshop capabilities of GUI agents. The benchmark contains 600 samples categorized into easy, medium, and hard modes, accompanied by expert-annotated editing trajectories. Evaluation on SOTA MLLMs shows that most models struggle even with tasks in the easy category. The authors also conduct a human-in-the-loop study, showing that MLLMs can help novice users complete the tasks more accurately and efficiently.

**Strengths:**

- The proposed benchmark is novel and has practical value, as Photoshop (or, more generally, controllable image editing) represents a common real-world task.

- The benchmark is highly challenging (SOTA models perform poorly even in the easy category) leaving substantial room for future model development.

- PSBench is evaluated on a wide range of SOTA models, and the authors provide a solid error analysis of model failures.

**Weaknesses:**

- Limited details are provided about the 600 individual tasks. The paper mentions 16 categories (Appendix E), but per-category statistics are missing. The authors claim that the tasks are diverse, yet no evidence supports this. Showing the distribution of task counts across categories and difficulty levels (easy / medium / hard) would strengthen this claim.

- The evaluation functions lack human validation for GPT-4o as a vision-language judge. Since VLMs can hallucinate, it is unclear how trustworthy these evaluations are. Alignment with human evaluation (even on a subset) is needed.

- The proposed metric NDEC, which compares the agent trajectory to the gold trajectory, seems oversimplified. While comparing two trajectories is difficult, a six-criterion checklist is too coarse-grained. The evaluation protocol is also vague, as it relies on an expert’s subjective judgment (e.g., whether a feature like Smart Objects is “used properly”). A clearer, more fine-grained evaluation protocol would improve reproducibility and soundness.

**Questions:**

**Question**

- In Table 2, the time horizon is defined as “the number of UI actions per task, reported as the average operation length for hard tasks.”
Why not report the average operation length across all tasks (easy, medium, and hard)? The current number might be inflated, making the comparison unfair.

**Comments on Writing**

- Line 292: “NDEC metric” is mentioned, but its formal definition appears only at line 303.

- Lines 399–404: “All models achieve overall NDEC scores above 70%, indicating that their generated action sequences not only accomplish the intended edits but also largely adhere to nondestructive editing principles.”
 However, according to Table 3, the models do not seem to accomplish the intended edits.

---

> ### Author Response · Authors · 2025-11-26
> **Response to reviewer TMYi(1/2)**
>
> >Weakness 1: Limited details are provided about the 600 individual tasks. The paper mentions 16 categories (Appendix E), but per-category statistics are missing. The authors claim that the tasks are diverse, yet no evidence supports this. Showing the distribution of task counts across categories and difficulty levels (easy / medium / hard) would strengthen this claim.
>
> We thank the reviewer for the valuable comments on task diversity and difficulty design. In response, we have added a new section (C.4) in the Appendix, which provides more comprehensive statistics, and we further clarify the design of task difficulty.
>
> First, the 16 types of image editing mentioned in the paper (Appendix E) are used to describe the real-world scenarios from which tasks are drawn. The difficulty classification, however, is not based on these 16 content types, but rather on the number of operation categories involved in a task. As shown in Appendix C.4.2, we systematically decomposed Photoshop’s complete operation system and summarized it into 6 major categories comprising 74 basic operations. Based on this, we adopted the following rigorous and reproducible difficulty classification:
> - Easy: tasks involve only 1 operation category
> - Medium: tasks involve a combination of 2–3 operation categories
> - Hard: tasks involve complex combinations spanning more than 3 categories
>
> This design ensures that difficulty is determined by the complexity of the operation structure rather than by subjective assessment of the “professionalism” of task content.
>
> For Medium tasks, operations are usually simple sequential combinations of different categories (e.g., applying a filter followed by a geometric transformation), intended to evaluate whether the agent can maintain stable planning when multiple tools appear simultaneously.
> In contrast, Hard tasks are not simple stacks of operations but are fully drawn from real-world professional image editing workflows. We carefully selected tasks from the most-viewed Photoshop tutorials on YouTube to ensure completeness, realism, and executability. Successfully completing these tasks requires not only cross-category tool proficiency but also understanding of professional editing logic, layer structure organization, and non-destructive workflow principles.
>
> For example, the Hard task “add glowing effects to the kangaroo” involves:
> - Selection and edge refinement (Select and Mask, Refine Edge, etc.)
> - Smart objects and layer management (Smart Object, grouping, etc.)
> - Overall tone and color adjustment (Color Lookup, Hue/Saturation, Levels)
> - Multi-level blur stacking for glow effects (Gaussian Blur combined with Linear Dodge)
> - Local color enhancement (Hue/Saturation with clipping mask)
>
> Although this task involves 5 operation categories, simply knowing these categories is far from sufficient. The agent must understand the complete workflow, tool-combination logic, and layer semantics.
>
> Therefore, designing difficulty based on combinations of operation categories better reflects the overall editing capability of GUI agents in professional software environments, compared to classification by content theme. Additionally, in Figure 7, we show the distribution of different operations across tasks of varying difficulty. We hope that Section C.4 helps the reviewer better understand our benchmark’s task design.
>
> >Weakness 2: The evaluation functions lack human validation for GPT-4o as a vision-language judge. Since VLMs can hallucinate, it is unclear how trustworthy these evaluations are. Alignment with human evaluation (even on a subset) is needed.
>
> To address the reviewer’s concern regarding the reliability of GPT-4o as a vision–language evaluation tool, we conducted a validation experiment on a subset of PSBench. Specifically, we randomly selected 150 successfully completed tasks, each of which was independently evaluated by both a human assessment group (three evaluators) and GPT-4o. The human group determined task success through majority voting, and the results were then compared with GPT-4o’s judgments.
>
> The results show that GPT-4o generally adopts a stricter evaluation stance. Each task was supplied with a target image edited by Photoshop experts to serve as a clear reference, which we believe leads GPT-4o to assign comparatively harsher scores. The computed Cohen’s κ coefficient is 0.71.
>
> Note that Cohen’s κ ranges from –1 to +1:
> - κ ≤ 0: agreement no better than chance;
> - 0–0.2: slight;
> - 0.2–0.4: fair;
> - 0.4–0.6: moderate;
> - 0.6–0.8: substantial;
> - 0.8: almost perfect.
>
> Thus, κ = 0.71 falls in the “substantial agreement” band, indicating strong alignment between GPT-4o and human evaluators. Despite the potential for hallucinations in vision–language models, this level of consistency supports the use of GPT-4o as a reliable evaluator for PSBench tasks.

---

> ### Author Response · Authors · 2025-11-26
> **Response to reviewer TMYi(2/2)**
>
> >Weakness 3： The proposed metric NDEC, which compares the agent trajectory to the gold trajectory, seems oversimplified. While comparing two trajectories is difficult, a six-criterion checklist is too coarse-grained. The evaluation protocol is also vague, as it relies on an expert’s subjective judgment (e.g., whether a feature like Smart Objects is “used properly”). A clearer, more fine-grained evaluation protocol would improve reproducibility and soundness.
>
> We thank the reviewer for the valuable comments on the NDEC metric. Here, we provide a clearer explanation of its design rationale, the role of the gold trajectory, and the consistency of its evaluation.
>
> First, the six checklist items in NDEC are directly derived from Adobe’s official guidelines on non-destructive editing (https://helpx.adobe.com/cn/photoshop/using/nondestructive-editing.html ). In professional Photoshop workflows, non-destructive principles primarily rely on six key mechanisms: Smart Objects, layer and filter masks, Smart Filters, Adjustment Layers, Duplicate Layers, and Blank Layers. Therefore, NDEC’s six items comprehensively cover the core non-destructive strategies defined in the official documentation, providing both a clear theoretical basis and grounding in industry practice.
>
> Second, we clarify the role of the gold trajectory in the evaluation framework. The gold trajectory is not intended to require the model to “mimic a path,” nor is it a reference for step-by-step comparison. Instead, it serves to define the semantic boundaries and evaluation criteria for the six NDEC items. Since Photoshop tasks exhibit high trajectory diversity (the same editing goal can often be achieved via multiple completely different feasible sequences of actions), we do not evaluate whether the model replicates the GT steps. Rather, the gold trajectory specifies which operations in a given task must be performed in a non-destructive manner.
>
> This approach ensures that even if different models follow entirely different operation paths, we can still assess whether their trajectories adhere to non-destructive editing principles based on a unified, explicit, and professional standard, thereby guaranteeing the objectivity, stability, and reproducibility of NDEC.
>
> Regarding the reviewer’s concern that “evaluation relies on expert judgment and may lack granularity,” we validated NDEC’s reliability through an annotation consistency experiment. Four independent annotators scored NDEC on 50 randomly sampled task results, and Fleiss’ Kappa was computed, yielding an average κ = 0.72, which falls within the “substantial agreement” range. This demonstrates that NDEC’s six checklist items exhibit strong inter-annotator consistency in practice and do not overly depend on subjective judgment of individual evaluators.
>
> > Question: In Table 2, the time horizon is defined as “the number of UI actions per task, reported as the average operation length for hard tasks.” Why not report the average operation length across all tasks (easy, medium, and hard)? The current number might be inflated, making the comparison unfair.
>
> We thank the reviewer for the suggestion. We agree that the average action length should be reported across all difficulty levels (easy, medium, hard) to provide a more comprehensive and fair comparison. Accordingly, we have updated Table 2 to report the average action length across all tasks, rather than only using statistics from the hard tasks.
>
> >Comments on Writing
>
> Thank you for your suggestion,we have corrected these two errors.

---

### Official Review · Reviewer_CQqw · 2025-10-28

**Soundness:** 3
**Presentation:** 3
**Contribution:** 3
**Rating:** 6
**Confidence:** 4

**Summary:**

The paper introduces the first benchmark for image editing within the Adobe Photoshop environment, emphasizing its core concept of non-destructive layer-based editing. Experiments show that leading MLLMs (e.g., Qwen2.5-VL, GPT-5, Gemini-2.5-Pro) achieve low task success rates but exhibit strong planning abilities. Moreover, human-in-the-loop experiments demonstrate that MLLMs can effectively assist novice users, significantly improving task success and reducing operation time.

**Strengths:**

The paper is well-structured and easy to follow, with a clear presentation and well-designed experiments. A well-designed benchmark for image editing is much needed to systematically evaluate the capabilities of agents in this domain.

**Weaknesses:**

1. The paper lacks a clear definition of task difficulty levels in Photoshop. Please clarify whether the difficulty is determined by the number of operation steps or by the intrinsic complexity of each operation. For example, “cropping” and “applying filters” clearly represent different types of tasks, and the criteria for difficulty classification should be explicitly defined. In addition, please provide a detailed statistics and categorization of all Photoshop operation types included in the study to justify the task design better.

2. Since the trajectory in Photoshop is not a fixed path, please further explain how the model handles diverse operation sequences and how the performance is evaluated under such variability.

3. The current MLLM model accuracy is quite low (up to only 18%), while end-to-end editing models demonstrate significantly better performance. Although the authors mention that the proposed method can effectively assist novice users in the human-in-the-loop experiments, the related results are not sufficiently demonstrated. It is recommended to further analyze the causes of low accuracy and to discuss whether the benchmark dataset used in this work is truly effective and representative.

4. Regarding the human-in-the-loop experiment, please specify the number and background of participants. The scale and diversity of participants play a crucial role in validating the effectiveness of the benchmark.

**Questions:**

It is strongly suggested to include a complete visualization of the agent workflow to help readers better understand the overall system mechanism and benchmark datasets. Specifically, please consider:
Visualizing the agent’s input and output at each step;
Presenting intermediate results along with their corresponding evaluation metrics.

---

> ### Author Response · Authors · 2025-11-26
> **Response to reviewer  CQqw(1/5)**
>
> Thank you for the detailed and constructive feedback! We treasure the opportunity to address your concerns and improve our work.
>
> > Weakness 1: The paper lacks a clear definition of task difficulty levels in Photoshop. Please clarify whether the difficulty is determined by the number of operation steps or by the intrinsic complexity of each operation. For example, “cropping” and “applying filters” clearly represent different types of tasks, and the criteria for difficulty classification should be explicitly defined. In addition, please provide a detailed statistics and categorization of all Photoshop operation types included in the study to justify the task design better.
>
> We thank the reviewer for the valuable suggestion. As you correctly pointed out, different Photoshop operations (such as cropping vs. applying filters) indeed have different inherent levels of professional difficulty. However, in the context of evaluating GUI Agents, these tasks can be uniformly abstracted into the following operational structure: locating the relevant function → configuring parameters → applying the operation. Therefore, the difficulty levels in our benchmark are not defined based on the professional threshold of any specific Photoshop feature, but rather on the combinatorial complexity determined by the number of required operation steps and the number of operation categories involved.
>
> To address your concern, we also include a new section (C.4.2) in the Appendix. As shown in Appendix C.4.2, we begin by systematically decomposing Photoshop’s operation system, categorizing it into 6 major groups comprising 74 atomic operations. Based on this structure, we adopt the following difficulty definitions:
> - Easy: involves only 1 operation category;
> - Medium: involves a combination of 2–3 operation categories;
> - Hard: involves more than 3 operation categories with complex interactions.
>
> In Medium tasks, operations are usually simple sequences across categories (e.g., applying a filter followed by a geometric transform). These tasks evaluate whether the agent can maintain stable planning ability when multiple tool types appear simultaneously. Hard tasks, in contrast, are not merely stacks of operations; they are sourced directly from real professional scenarios—we selected tasks from the most-viewed professional Photoshop tutorials on YouTube to ensure completeness, realism, and practical relevance. Completing Hard tasks requires not only planning across multiple operation categories but also an understanding of professional image editing workflows.
>
> For example, in the Hard task “Add glowing effects to the kangaroo,” the full workflow includes:
> - Selection and cutout (Selection tools, Select and Mask, Refine Edge, etc.)
> - Layer management and Smart Objects (Smart Object creation, group organization, etc.)
> - Tonal and atmosphere adjustments (Color Lookup, Hue/Saturation, Levels, etc.)
> - Glow construction using filters (Gaussian Blur with multiple linear-dodge layers)
> - Local color enhancement (Hue/Saturation with Clipping Masks)
>
> This task covers 5 operation categories. However, merely knowing these operations is insufficient—the agent must also understand the overall professional editing workflow, including nondestructive editing principles, layer structure organization, and light-effect construction logic. Thus, defining difficulty through combinations of operation categories better reflects a GUI Agent’s comprehensive capabilities in real software environments, rather than its proficiency with isolated operations.
>
> We fully acknowledge that simple operations like cropping and certain advanced filters have different skill thresholds in professional usage. However, relying solely on such “single-operation difficulty differences” would not meaningfully reveal a GUI Agent’s real decision-making and execution capabilities within Photoshop. By contrast, a difficulty definition based on combinatorial complexity of operation categories aligns more closely with the capability modeling of GUI Agents and more effectively differentiates their performance in multi-step, multi-stage, and multi-tool collaborative workflows.
>
> To further demonstrate the validity and coverage of our task design, Appendix C.4 provides complete statistics, including:
> - the 6 major operation categories covered by the benchmark (74 atomic operations in total),
> - and the 16 professional editing workflows included (such as compositing, special effects creation, skin retouching, etc.).
>
> Together, these design components ensure that our task system has a clear and well-founded difficulty structure within the Photoshop domain and provides broad and representative coverage across both the operation dimension and the professional workflow dimension, thereby offering a solid foundation for evaluating GUI Agents’ comprehensive editing abilities in real-world software environments.

---

> ### Author Response · Authors · 2025-11-26
> **Response to reviewer  CQqw(2/5)**
>
> >Weakness 2: Since the trajectory in Photoshop is not a fixed path, please further explain how the model handles diverse operation sequences and how the performance is evaluated under such variability.
>
> We thank the reviewer for the insightful question. As you correctly pointed out, in Photoshop, a single task can typically be completed through multiple possible operation paths, and such high trajectory diversity is indeed a fundamental characteristic of professional image-editing software. Therefore, from the very beginning, our benchmark was designed to allow the model to adopt any feasible operation sequence, rather than constraining it to one specific path.
>
> Our evaluation framework consists of two complementary dimensions:
> 1. Success rate, which is determined solely by the final edited result.
>  Regardless of the interface path or sequence of actions taken, as long as the final output passes the task-specific evaluation function, it is counted as a success. This outcome-based evaluation naturally accommodates multi-path strategies, allowing diverse exploration behaviors to be accepted.
> 2. NDEC (Non-Destructive Editing Consistency), which evaluates whether the model follows professional editing paradigms, independent of the exact path taken.
>  Photoshop’s nondestructive editing principles have well-defined criteria, such as the use of adjustment layers, smart filters, masks, duplicate layers, etc. NDEC checks the model’s actual trajectory step by step, but it does not require the model to imitate the gold trajectory, nor does it penalize alternative valid paths.
> Within this framework, the purpose of the gold trajectory is not to constrain the agent’s path, but rather to serve as a professional reference anchor that defines the semantics and boundaries of NDEC:
> - It specifies which steps must be nondestructive for a given task;
> - It provides expert-level criteria for each checklist item;
> - It enables us to determine whether the model’s chosen path adheres to professional editing principles, rather than whether it matches the gold trajectory itself.
>
> In other words, the gold trajectory provides professional standards and operational norms, and NDEC checks whether the model follows these principles along its self-chosen path. If the model adopts a nondestructive workflow, NDEC will reflect this positively; if it uses destructive methods, NDEC will clearly deduct points, regardless of the specific path taken.
>
> To further illustrate how a GUI Agent explores and selects its own UI interaction paths, we have added a new section (C.5) in the Appendix. This section demonstrates that the agent can freely plan within a large action space, while our evaluation framework remains robust and fair in handling such trajectory diversity.

---

> ### Author Response · Authors · 2025-11-26
> **Response to reviewer  CQqw(3/5)**
>
> >Weakness 3: The current MLLM model accuracy is quite low (up to only 18%), while end-to-end editing models demonstrate significantly better performance. Although the authors mention that the proposed method can effectively assist novice users in the human-in-the-loop experiments, the related results are not sufficiently demonstrated. It is recommended to further analyze the causes of low accuracy and to discuss whether the benchmark dataset used in this work is truly effective and representative.
>
> First, we agree with the reviewer’s observation that current MLLMs exhibit relatively low task success rates in professional software environments. Tasks in PSBench inherently require highly precise UI grounding, complex software state understanding, and fine-grained sequential action planning. Although existing general-purpose MLLMs excel in visual comprehension and reasoning, they still show substantial limitations in cross-state UI grounding, cross-window information alignment, parameter-level slider manipulation, state tracking under UI occlusion, and multi-step dependency reasoning within Photoshop. These capability gaps are the primary causes of the low success rates.
>
> To more comprehensively analyze the limitations of GUI agents in a professional, large-scale image editing application like Photoshop, we have added five groups of additional experiments, including: one latest general-purpose model, Gemini-3-Pro-Preview; two specialized models, UI-TARS-2-2509 and OpenCUA-7B; and two agentic frameworks, Agent S3 w/ GPT-5 and GTA1 w/ GPT-5.
>
> The results are as follows:
>
> | GUI Agent           | Easy SR (LR/NLR) | Medium SR (LR/NLR) | Hard SR (LR/NLR) | Overall SR (LR/NLR) |
> |---------------------|------------------|--------------------|------------------|---------------------|
> | gemini-3-pro-preview | 13.08 % / 13.98 % | 0.00 % / 6.25 %   | 0.00 % / 0.00 %  | 2.95 % / 11.90 %   |
> | UI-TARS-2-2509       | 7.48 % / 39.78 %  | 2.38 % / 28.13 %  | 4.52 % / 0.00 %  | 4.43 % / 36.51 %   |
> | opencua-7b           | 16.82 % / 45.16 % | 4.76 % / 34.38 %  | 3.52 % / 0.00 %  | 6.96 % / 42.06 %   |
> | agent s3 w/ GPT-5    | 41.12 % / 69.89 % | 25.00 % / 40.63 % | 18.09 % / 0.00 % | 25.74 % / 61.90 %  |
> | GTA1 w/ GPT-5        | 32.71 % / 58.06 % | 16.07 % / 37.50 % | 13.57 % / 0.00 % | 18.78 % / 52.38 %  |
>
> | GUI Agent           | NDEC (Easy) | NDEC (Medium) | NDEC (Hard) | NDEC (All) |
> |---------------------|-------------|---------------|-------------|------------|
> | gemini-3-pro-preview | 91.00 %     | 84.67 %       | 64.00 %     | 79.89 %    |
> | UI-TARS-2-2509       | 74.33 %     | 68.00 %       | 47.33 %     | 63.22 %    |
> | opencua-7b           | 73.00 %     | 66.67 %       | 45.67 %     | 61.78 %    |
> | agent s3 w/ GPT-5    | 89.00 %     | 83.67 %       | 63.33 %     | 78.67 %    |
> | GTA1 w/ GPT-5        | 88.33 %     | 83.67 %       | 63.33 %     | 78.44 %    |
>
> We observe that, compared to GUI agents built on general-purpose MLLMs, specialized models indeed achieve noticeably higher task success rates. However, their NDEC scores consistently decline, indicating a degradation in their professional editing competence and adherence to non-destructive editing principles within Photoshop. In contrast, existing agentic frameworks adopt a more reasonable division of labor: using a strong planning-capable MLLM such as GPT-5 as the high-level planner, while employing a model with strong UI grounding capabilities (e.g., UITARS) as the executor. This hybrid architecture effectively combines the strengths of both types of models. The experimental results show that such frameworks not only significantly outperform single-model solutions in task success rate, but also maintain a high level of professional editing quality and compliance with non-destructive principles. Nevertheless, even the current best-performing Agent S3 achieves only an 18.09% success rate on Hard tasks, highlighting that operating within complex professional software environments remains a substantial challenge for current GUI agents.
>
> (Continued below)

---

> ### Author Response · Authors · 2025-11-26
> **Response to reviewer  CQqw(4/5)**
>
> (Continued)
>
> Through further analysis of failure cases, we find that current GUI agents still exhibit clear structural limitations when operating in the Photoshop environment:
>
> A. Insufficient understanding of professional workflows and user practices in Photoshop
> - The models lack a deep understanding of the software itself, particularly regarding shortcut usage. Photoshop contains many professional shortcuts that greatly improve efficiency. For example, in the “Convert the image to grayscale” task, experienced users would directly use Ctrl + Shift + U, completing the task in one step, while the model still expands the menu layer by layer, which is slow and unprofessional. Similarly, the model mistakenly assumes that Ctrl + A clears the content of a numerical input field, while in Photoshop this shortcut selects the entire pixel area. As a result, the model fails to clear the value and subsequently produces out-of-range inputs (e.g., turning “30” into “3050”).
> - The model also shows misunderstandings of certain professional terms. For example, it misinterprets the portrait-retouching concept of “highlights” as “lighting,” and in a task related to enhancing facial contours, it incorrectly adds orange light spots instead of using the professional “Curves adjustment + mask painting” method.
> - The model further lacks mastery of advanced editing techniques common among Photoshop professionals. For instance, in the “add glowing effects to the windows and street lights in the image” task, instead of using paths, masks, blending modes, and Gaussian blur to create realistic glows, it simply paints with a brush—leading to unnatural and unprofessional results.
>
> B. Insufficient UI grounding accuracy
> - Although trained on large-scale UI datasets, the models frequently mis-locate UI elements in complex interfaces. For example, even when the planner correctly specifies clicking the “Filter” menu, the model repeatedly clicks “Select” or “3D,” showing a clear spatial offset (“Filter” is positioned between them).
> - Inaccurate grounding further compromises the model’s ability to perform fine-grained spatial reasoning. While the model can complete coarse tasks such as “select the center of the image,” it completely fails on tasks requiring precise region identification, such as “select a specific tree in the image.”
>
> C. Lack of aesthetic awareness and visual coherence
> - The model’s editing behavior is mechanical and lacks basic photographic and post-processing aesthetic knowledge. For example, when adding a soft orange glow, it directly paints with an orange brush without considering light direction, perspective, surface reflectance, or overall lighting consistency, resulting in outputs that do not meet professional aesthetic standards. (Under our evaluation criteria, such aesthetic deviations are not counted as task failures.)
>
> > Weakness 4: Regarding the human-in-the-loop experiment, please specify the number and background of participants. The scale and diversity of participants play a crucial role in validating the effectiveness of the benchmark.
>
> In our human-in-the-loop experiment, we recruited 24 undergraduate students majoring in computer-related disciplines. All participants possessed basic software operation skills but were complete novices in Photoshop: each reported a total usage time of less than two hours and had not received any form of image-editing training.
>
> We added a new section (D.4) in the appendix that provides more details about our human–machine experiments. The motivation for this experiment comes from one of our key findings: although current general-purpose MLLMs still suffer from insufficient UI grounding capabilities, they exhibit near expert-level competence in planning non-destructive professional workflows. Based on this observation, we designed a GUI Assistant mode to explore whether human–AI collaboration can compensate for the grounding bottleneck.
>
> Our human-in-the-loop experiment clearly demonstrates the effectiveness of this approach, particularly in terms of efficiency gains. By adopting a complementary mechanism—the model handles planning while the user handles execution—this mode effectively mitigates the MLLM’s grounding weaknesses. Users can obtain directly executable, professional-grade editing instructions without needing to search for, compare, or understand complex tutorials.
>
> The results show improvements in both success rate and completion time across all difficulty levels. Notably, in high-difficulty tasks, the collaboration mode achieves higher success rates while significantly reducing operation time compared to other conditions, showcasing a unique efficiency advantage.

---

> ### Author Response · Authors · 2025-11-26
> **Response to reviewer  CQqw(5/5)**
>
> >Question: It is strongly suggested to include a complete visualization of the agent workflow to help readers better understand the overall system mechanism and benchmark datasets. Specifically, please consider: Visualizing the agent’s input and output at each step; Presenting intermediate results along with their corresponding evaluation metrics.
>
> We appreciate your suggestion. We provide the following clarification regarding the agent workflow:
> - Initial input to the GUI Agent:
>   - The agent receives two components as input at Step 1: (1) the task instruction, and (2) the initial screenshot, which includes both the unedited source image and the full Photoshop interface.
> - Inputs for subsequent steps (Step n, for all n ≥ 2):
>   - Each step receives:
>     - The updated screenshot obtained after executing the action from Step (n − 1)
>     - The accumulated memory from the previous (n − 1) steps, which stores the agent’s intermediate reasoning, state analysis, and action planning.
>
> As shown in Table 7, the workflow is decomposed into four synchronized components:
> - Step n: indicates the current reasoning step;
> - Screenshot: provides the interface state after executing Step (n − 1);
> - Agent Response: contains the model’s analysis of the current state and justification for the next action;
> - Action Code: is the executable Python code issued by the agent to interact with Photoshop.
>
> Overall, the screenshot produced after each executed action becomes the visual input for the next reasoning step, forming a closed-loop, step-by-step interaction cycle. This design ensures that the GUI Agent maintains coherent reasoning, stable grounding, and controlled execution throughout multi-step editing tasks.
>
> After the task is completed, the benchmark automatically saves the edited result and invokes the evaluation function to assess its quality. At the same time, it extracts the agent’s internal action trajectory from the GUI Agent’s responses. These trajectories are then used for checklist-based NDEC evaluation to determine whether the GUI Agent adhered to Photoshop’s non-destructive editing principles during the task.
>
> To make this workflow more intuitive, we provide a task example in Appendix C.5.

---

### Official Review · Reviewer_SVrh · 2025-10-30

**Soundness:** 2
**Presentation:** 2
**Contribution:** 2
**Rating:** 4
**Confidence:** 3

**Summary:**

This paper introduces PSBench, the first specialized benchmark for evaluating GUI agents in Adobe Photoshop, a professional image editing application. The benchmark comprises 600 human-annotated tasks divided into three difficulty levels (Easy, Medium, Hard), each with 200 tasks covering both layer-related and non-layer-related operations. Tasks range from simple atomic operations to complex real-world workflows derived from popular YouTube tutorials. Each task includes an original image, target output, and expert-annotated "gold trajectory" demonstrating non-destructive editing practices. The authors propose a novel evaluation metric, Non-Destructive Editing Consistency (NDEC), which measures adherence to professional Photoshop workflows using a 6-criteria checklist. Comprehensive experiments on seven state-of-the-art MLLMs (GPT-4o, GPT-5, Claude-4-Sonnet, Gemini-2.5-Pro, etc.) reveal extremely low success rates—the best model (GPT-4o) achieves only 17.46% on non-layer tasks and 3.80% on layer-related tasks. In contrast, end-to-end image editing models achieve 90-96% overall success. Human-in-the-loop experiments demonstrate that AI assistance significantly improves novice user performance, suggesting collaborative approaches as a promising direction.

**Strengths:**

1、The benchmark addresses a significant gap in GUI agent evaluation by targeting professional software with complex interfaces and long-horizon tasks, moving beyond existing web and OS benchmarks. The focus on Photoshop is highly relevant given its widespread use in professional image editing.

2、The Non-Destructive Editing Consistency metric is a valuable contribution that goes beyond simple outcome-based evaluation to assess workflow quality and professional practice adherence. This process-oriented metric is particularly appropriate for complex software domains.

**Weaknesses:**

1、 With only 600 tasks total (200 per difficulty level), the benchmark is relatively small compared to modern datasets. The task distribution is limited to specific Photoshop operations and may not cover the full breadth of professional editing workflows. No discussion of task diversity metrics or coverage analysis is provided

2、The NDEC metric, while innovative, is based on a fixed 6-criteria checklist that may not capture all aspects of professional practice. The paper does not validate whether these 6 criteria comprehensively represent non-destructive workflows or discuss inter-annotator agreement on NDEC scoring. The binary checklist approach may miss nuanced differences in workflow quality.

3、 The paper does not evaluate specialized GUI agent models (AutoGLM-OS-9B, OpenCUA-32B, UITARS) or agent frameworks (CoACT-1) that reportedly outperform general-purpose MLLMs on OSWorld.

4、Comparing GUI agents with end-to-end image editing models (Table 6) is somewhat misleading since these models solve the task through completely different mechanisms (direct image generation vs. software manipulation). The 90%+ success rates of end-to-end models primarily demonstrate that PSBench tasks can be solved via image generation, but this doesn't provide actionable insights for GUI agent development.

**Questions:**

see Weaknesses

---

> ### Author Response · Authors · 2025-11-26
> **Response to reviewer SVrh(1/4)**
>
> Thank you for the detailed and constructive feedback! We treasure the opportunity to address your concerns and improve our work.
>
> > Weakness 1: With only 600 tasks total (200 per difficulty level), the benchmark is relatively small compared to modern datasets.
>
> Regarding the concern of “insufficient task quantity and limited coverage,” we would like to clarify the following: although our benchmark contains only 600 tasks, this scale is mainly determined by the inherent complexity of Photoshop itself, whose data collection cost is significantly higher than that of office software or web-based environments.
>
> First, when constructing each task, we require annotators to fully execute the entire editing workflow in a real Photoshop environment according to the instruction, in order to produce a reference result for evaluation. According to our statistics, an easy task takes an average of about 5 minutes, a medium task about 10 minutes, and a hard task typically more than 20 minutes (Figure 2 provides the detailed human cost). Currently, for such a complex image-editing software, there is no automatic method that can replace manual work while maintaining high quality.
>
> Second, for tasks involving layers or non-destructive editing, we need to provide manually written gold trajectories. Although some prior work has attempted to use RPA tools and MLLMs for semi-automatic collection (e.g., GUI-Robust [1]), these methods cannot ensure the strict non-destructive editing standards required by Photoshop workflows. Therefore, we require annotators to manually construct the trajectories by strictly following Adobe’s official Photoshop documentation. In cases where ambiguities or uncertainties arise, two independent annotators first provide their own interpretations, and if discrepancies occur, a third annotator is involved to offer an additional judgment. The annotators then discuss and reach a consensus on the final trajectory, ensuring its accuracy, clarity, and consistency.
>
> Third, we wrote a total of 377 evaluation functions, and validated them individually, which further increases the data construction cost. Moreover, unlike many closed-environment benchmarks (such as AITW [2], WebShop [3]) that rely on large-scale user logs or synthetic data—which do not require actually performing tasks and thus are easier to scale—our benchmark is an open-environment benchmark. It requires agents to complete editing operations end-to-end in the real Photoshop software, making it impossible to adopt the same data expansion strategies, while more faithfully measuring the agent’s operational capability. In fact, the most authoritative open-environment benchmark to date, OSWorld [4], contains only 369 tasks, which demonstrates that building large-scale, high-quality task sets in real software environments is intrinsically very challenging.
>
> [1]: GUI-Robust: A Comprehensive Dataset for Testing GUI Agent Robustness in Real-World Anomalies https://arxiv.org/abs/2506.14477
> [2]: Android in the Wild: A Large-Scale Dataset for Android Device Control: https://arxiv.org/abs/2307.10088
> [3]: WebShop: Towards Scalable Real-World Web Interaction with Grounded Language Agents: https://arxiv.org/abs/2207.01206
> [4]: OSWorld: Benchmarking Multimodal Agents for Open-Ended Tasks in Real Computer Environments: https://arxiv.org/abs/2404.07972
>
> > Weakness 2: The task distribution is limited to specific Photoshop operations and may not cover the full breadth of professional editing workflows. No discussion of task diversity metrics or coverage analysis is provided.
>
> To address your concern, we further include a new section (C.4) in the Appendix. In this section, we present detailed statistics on task distribution, showing that the benchmark covers:
>
> - **6 major categories** (such as *Geometric Transformations*, *Filter Effects*, etc.),
> - **74 types of Photoshop operations** (such as *Crop*, *Curves*, *Lasso Tool*, etc.),
> - **16 types of professional editing workflows** (such as *compositing*, *color grading*, *special effects creation*, etc.).
>
> These components together ensure that the task set is sufficiently representative and diverse within the Photoshop domain.

---

> ### Author Response · Authors · 2025-11-26
> **Response to reviewer SVrh(2/4)**
>
> >Weakness 3: The NDEC metric, while innovative, is based on a fixed 6-criteria checklist that may not capture all aspects of professional practice. The paper does not validate whether these 6 criteria comprehensively represent non-destructive workflows or discuss inter-annotator agreement on NDEC scoring. The binary checklist approach may miss nuanced differences in workflow quality.
>
> We thank the reviewer for the valuable comments on the NDEC metric. Our six inspection criteria are directly based on Adobe’s official documentation on non-destructive editing (https://helpx.adobe.com/cn/photoshop/using/nondestructive-editing.html). In professional Photoshop editing workflows, non-destructive principles are mainly reflected through the use of mechanisms such as Smart Objects, masks (including layer masks and filter masks), Smart Filters, adjustment layers, duplicate layers, and blank layers. Therefore, the six checklist items proposed in the paper (Proper use of Smart Objects, Masks, Smart Filters, Adjustment Layers, Duplicate Layers, and Blank Layers) cover the major non-destructive editing strategies defined in the official documentation, and have clear theoretical grounding and professional practice foundations.
>
> To verify the reliability of the NDEC metric, we conducted an inter-annotator consistency experiment: four annotators independently scored 50 randomly selected tasks, and Fleiss’ Kappa was computed for each checklist item. The average κ = 0.72 was obtained.
> It should be noted that Fleiss’ Kappa ranges from –1 to +1:
> - κ ≤ 0: agreement worse than chance;
> - 0 &lt; κ ≤ 0.2: slight agreement;
> - 0.2 &lt; κ ≤ 0.4: fair agreement;
> - 0.4 &lt; κ ≤ 0.6: moderate agreement;
> - 0.6 &lt; κ ≤ 0.8: substantial agreement;
> - κ &gt; 0.8: almost perfect agreement.
>
> Thus, κ = 0.72 falls in the “substantial agreement” band, indicating that NDEC achieves good reproducibility and an acceptable level of subjective-bias control in practical annotation.
> Regarding the reviewer’s comment that “a binary checklist may fail to capture fine-grained differences in workflow quality,” we partially agree. Indeed, in certain editing workflows, there may be cases of “partial compliance” or “suboptimal but not entirely incorrect” operations, which cannot be reflected by a 0/1 scoring scheme. However, we adopt a binary checklist because many operations at the core of Photoshop’s non-destructive principles are essentially binary behaviors, such as:
> - Whether edits are performed on separate layers (non-destructive) versus directly modifying the background layer (destructive)
> - Whether filters are applied as Smart Filters (non-destructive) versus applied directly in normal mode (destructive)
>
> Thus, from a practical perspective, the goal of NDEC is not to characterize continuous editing quality, but rather to assess whether the agent adheres to the key structural principles of professional editing pipelines, and binary scoring is sufficient to reflect this.
>
> Nonetheless, we acknowledge the reviewer’s point that “a more fine-grained metric may be more expressive.” In future versions, we will further explore adding multi-level scoring or weighting schemes to more comprehensively capture subtle differences in editing quality, such as adopting a three-level scoring system of 0 (non-compliant), 1 (partially compliant), and 2 (fully compliant), or introducing weight mechanisms based on the importance of operations to improve the sensitivity and expressiveness of the metric.

---

> ### Author Response · Authors · 2025-11-26
> **Response to reviewer SVrh(3/4)**
>
> > Weakness 4: The paper does not evaluate specialized GUI agent models (AutoGLM-OS-9B, OpenCUA-32B, UITARS) or agent frameworks (CoACT-1) that reportedly outperform general-purpose MLLMs on OSWorld.
>
> We thank the reviewer for the valuable suggestions. Based on your feedback, we have added five additional experiments:
> - **Gemini-3-Pro-Preview** (latest general-purpose model)
> - **UI-TARS-2-2509** & **OpenCUA-7B** (specialized GUI models)
> - **Agent S3 w/ GPT-5** & **GTA1 w/ GPT-5** (agentic frameworks)
>
>  The results are as follows:
>
> | GUI Agent           | Easy SR (LR/NLR) | Medium SR (LR/NLR) | Hard SR (LR/NLR) | Overall SR (LR/NLR) |
> |---------------------|------------------|--------------------|------------------|---------------------|
> | gemini-3-pro-preview | 13.08 % / 13.98 % | 0.00 % / 6.25 %   | 0.00 % / 0.00 %  | 2.95 % / 11.90 %   |
> | UI-TARS-2-2509       | 7.48 % / 39.78 %  | 2.38 % / 28.13 %  | 4.52 % / 0.00 %  | 4.43 % / 36.51 %   |
> | opencua-7b           | 16.82 % / 45.16 % | 4.76 % / 34.38 %  | 3.52 % / 0.00 %  | 6.96 % / 42.06 %   |
> | agent s3 w/ GPT-5    | 41.12 % / 69.89 % | 25.00 % / 40.63 % | 18.09 % / 0.00 % | 25.74 % / 61.90 %  |
> | GTA1 w/ GPT-5        | 32.71 % / 58.06 % | 16.07 % / 37.50 % | 13.57 % / 0.00 % | 18.78 % / 52.38 %  |
>
> | GUI Agent           | NDEC (Easy) | NDEC (Medium) | NDEC (Hard) | NDEC (All) |
> |---------------------|-------------|---------------|-------------|------------|
> | gemini-3-pro-preview | 91.00 %     | 84.67 %       | 64.00 %     | 79.89 %    |
> | UI-TARS-2-2509       | 74.33 %     | 68.00 %       | 47.33 %     | 63.22 %    |
> | opencua-7b           | 73.00 %     | 66.67 %       | 45.67 %     | 61.78 %    |
> | agent s3 w/ GPT-5    | 89.00 %     | 83.67 %       | 63.33 %     | 78.67 %    |
> | GTA1 w/ GPT-5        | 88.33 %     | 83.67 %       | 63.33 %     | 78.44 %    |
>
> We observe that, compared with GUI Agents built on general-purpose MLLMs, specialized models do achieve a clear improvement in task success rate, but their NDEC scores consistently drop. This indicates that their professional editing capability in Photoshop and adherence to non-destructive principles actually degrade. In contrast, existing agentic frameworks employ a more reasonable division-of-labor architecture: using strong-planning MLLMs such as GPT-5 as the high-level decision maker, while models like UITARS with strong UI grounding ability serve as the execution layer, thereby combining the strengths of both types of models. Experimental results show that such frameworks not only significantly outperform single-model approaches in terms of task success rate, but also maintain a high level of professionalism and adherence to non-destructive editing principles. However, even the best-performing system currently, Agent S3, achieves only 18.09% success on Hard tasks, indicating that GUI Agents still face considerable challenges in operating complex professional software.

---

> ### Author Response · Authors · 2025-11-26
> **Response to reviewer SVrh(4/4)**
>
> > Weakness 5：Comparing GUI agents with end-to-end image editing models (Table 6) is somewhat misleading since these models solve the task through completely different mechanisms (direct image generation vs. software manipulation). The 90%+ success rates of end-to-end models primarily demonstrate that PSBench tasks can be solved via image generation, but this doesn't provide actionable insights for GUI agent development.
>
> Yes, your understanding is completely correct. Our benchmark(PSBench)—including its task design, execution paradigm, and evaluation metrics—is entirely constructed to assess the capability of GUI Agents operating within the real Photoshop software environment. Therefore, directly comparing GUI agents with end-to-end image editing models does not meaningfully reflect the comparability of their abilities.
>
> The reason we include experiments with end-to-end image editing models in the paper is not to demonstrate their competitiveness in GUI operation, but rather to highlight an important phenomenon: although these models can generate images that visually “match the task description” with very high success rates, their results cannot substitute for professional Photoshop editing workflows.
> In Table 7, we present side-by-side comparisons between Photoshop’s real editing outputs and those generated by end-to-end image editing models. These examples clearly show that:
> - Image generation models often only approximately complete the task and lack the precise control required for professional editing.
> - Their results exhibit obvious gaps compared to true Photoshop edits, especially in fine-grained details and localized corrections.
> - Many tasks require layered, reversible editing and professional compositing conventions—capabilities completely beyond current end-to-end models.
>
> Therefore, the purpose of including this set of experiments is to emphasize that, despite rapid advances in image generation, the precision, controllability, and professional workflow offered by software like Photoshop remain irreplaceable. For this reason, developing GUI Agents capable of genuinely operating Photoshop continues to have substantial value.
>
> In other words, this experiment is intended to highlight the limitations of end-to-end models in terms of controllability and professional editing quality, rather than to position them as competing approaches to GUI Agents. We hope this clarifies our motivation for including this comparative study.

---

### Author Response · Authors · 2025-11-26
**General Response**

Dear Reviewers,

We sincerely thank all reviewers for their careful reading and insightful feedback on our work. Your comments have not only helped us further refine the paper but also provided valuable guidance for our future research directions. We especially appreciate the reviewers’ positive recognition of the significance of our benchmark. For example, Reviewer SVrh noted that PSBench, as the first benchmark specifically designed to evaluate GUI agents in Adobe Photoshop, a professional image editing application, effectively fills a critical gap in assessing GUI agents in professional-grade software, and highlighted the importance of the NDEC metric for evaluating workflow quality. Reviewer CQqw acknowledged the clarity of the paper’s structure and experimental design, emphasizing the urgent need for a systematic image-editing benchmark in the current research landscape. Reviewer TMYi underscored the practical relevance of the benchmark, its high level of difficulty, and the thorough analysis of SOTA model failures.

Our detailed responses address the specific questions and suggestions raised by each reviewer, covering several key aspects:
- Additional statistical details and task categorization:
 In the revised version, we have supplemented more comprehensive information, including six major categories of Photoshop tasks, a total of 74 fundamental operations, and 16 types of professional editing workflows covered in our benchmark.
- Further clarification of the NDEC metric:
 Reviewers SVrh and TMYi expressed concerns that the six-item fixed checklist may not capture all nuances of professional practice. In response, we elaborated on the design rationale of NDEC—fully grounded in Adobe’s official documentation on non-destructive editing—and additionally reported inter-annotator agreement results to demonstrate the reliability and reproducibility of NDEC scoring.
- Additional experiments and expanded error analysis:
 Reviewer SVrh noted the absence of evaluations on specialized models and agentic frameworks, and Reviewer CQqw suggested deeper failure analysis. To address this, we incorporated five additional groups of experiments and conducted a systematic failure analysis of these models.
- Reliability of GPT-4o as an evaluator:
 Reviewer TMYi raised concerns that VLMs may hallucinate. We therefore supplemented alignment studies comparing GPT-4o’s evaluations with human judgments, demonstrating the stability and reliability of GPT-4o as an evaluator in this setting.
- Additional details of human-in-the-loop experiments:
 Reviewer CQqw pointed out the lack of information about participant numbers and backgrounds in the human-in-the-loop experiments. We have included these details in the revised manuscript.

Furthermore, please allow us to reiterate the core contribution of our work: PSBench is the first benchmark specifically designed to evaluate GUI agents in Adobe Photoshop, a professional image editing application, and it innovatively introduces a new evaluation metric tailored to the characteristics of Photoshop—Non-Destructive Editing Consistency (NDEC). This metric allows the assessment of GUI agents to go beyond mere task success rates, further measuring models’ understanding of and adherence to professional software workflows. We believe that PSBench will provide meaningful momentum for advancing GUI agents in complex professional software environments and serve as a valuable reference benchmark for future research.

Once again, we deeply appreciate the time and expertise you have shared with us. Your encouraging feedback motivates us to continue advancing this work for the broader community, and we are more than happy to add clarifications to address any additional recommendations and reviews from you.

Best regards,
Authors of Paper 1921

---

### Meta-Review · Area_Chair_UBHC · 2025-12-29

**Summary:**

This paper introduces PSBench, the first benchmark dedicated to evaluating GUI agents in Adobe Photoshop environment, featuring 600 tasks and a novel Non-Destructive Editing Consistency (NDEC) metric. While reviewers acknowledged novelty and value of this benchmark, they raised concerns regarding the relatively small dataset size, the quality of the proposed metrics, lack of comparison with specialized GUI agent models and the unfair comparison with end-to-end models. Specifically, reviewers questioned the reliability of the fixed NDEC checklist, the trustworthiness of GPT-4o as an automated judge, and the absence of specialized GUI agent models in the evaluation.

**Reviewer Concerns:**

The authors provided a rebuttal to address most reviewers' concerns. They validated the NDEC metric and GPT-4o evaluator by demonstrating substantial agreement with human annotators (Kappa scores > 0.7), and they added evaluations for specialized models (e.g., UI-TARS, Agent S3) as requested. The authors also provided detailed statistics on task coverage and clarified definitions of difficulty. The limitation regarding the dataset size (600 tasks) remains, but the authors offered a compelling justification based on the high cost of collecting high-quality, non-destructive professional workflows.

**Reviewer Scores:**

Reviewer CQqw (Score 6) would likely strengthen their support to 7, given that the issues  regarding difficulty definitions and human-in-the-loop details are clearly explained. Reviewers SVrh and TMYi might marginally increase their scores from 4 to 5 due to the addition of specialized model baselines and GPT4o judge validation studies. However, it is also probable they would maintain the original scores as the limitations regarding the dataset size and the potential oversimplification of the evaluation metric are still not well explained.

---

### Decision · Program_Chairs · 2026-01-26

Reject